# Patients' experiences of living with low anterior resection syndrome three to six months after colorectal cancer surgery: A phenomenological study

Camilla Løvall[1], Lotte Miriam Eri Mjelde[2], Leslie S. P. Eide[3], Marit Hegg Reime[3,4]*

1 Department of Surgery, Vestfold Hospital Trust, Tønsberg, Norway, 2 Department of Gyneacology, Vestfold Hospital Trust, Tønsberg, Norway, 3 Department of Health and Caring Sciences, Western Norway University of Applied Sciences, Bergen, Norway, 4 Lovisenberg Diaconal University College, Oslo, Norway

* marit.hegg.reime@hvl.no

**Data Availability Statement:** The raw data used in this study cannot be shared publicly as it consists of participant interview transcripts. Participants

## Abstract

### Background

Increased use of sphincter-preserving surgery following colorectal cancer has led to more people living with low anterior resection syndrome (LARS), a disordered bowel function that significantly impacts quality of life. Little is known about how patients experience the first months following sphincter-preserving surgery and having LARS.

### Objective

To shed light on what it means to live with LARS in the first three to six months after colorectal cancer sphincter-preserving surgery.

### Method

A qualitative study with a phenomenological approach constitutes the study design. Semi-structured interviews were conducted with five participants from September 2022 to January 2023. The transcribed interviews were analysed using Giorgi's phenomenological method.

### Results

Five themes emerged from the analysis: bowel emptying becomes all-consuming and affects both daily life and working life, you hear what they're saying, but don't understand what it means until your body goes through it, low anterior resection syndrome may impact sexual life, leaving feelings of guilt, it doesn't matter what the circumstances are, but rather how one deals with them, and support and follow-up from healthcare professionals, employers, family and friends are crucial for living a good life with LARS.

### Conclusion

Participants described struggles living with major LARS in the early period following hospital discharge. However, few months later, they had developed strategies enabling them to

agreed to share their experiences on this sensitive topic, and provided consent to participation on the basis that potentially identifiable data would not be shared. This provision was approved by the Norwegian Agency for Shared Services in Education and Research (Reference number 849190). While information such as names, locations, and organizations were removed during transcriptions, participants may still be identifiable in the full transcripts given the study context, so it is not possible to share the raw study data. Therefore, only illustrative quotes from the transcripts have been included in this paper. However, all relevant data are within the paper. For more information about data requests relating to this study please contact the corresponding author Marit Hegg Reime (Marit.Hegg.Reime@hvl.no) or the Norwegian Agency for Shared Services in Education and Research (https://sikt.no/en/om-sikt/contact-us#Contactform).

**Funding:** The author(s) received no specific funding for this work.

**Competing interests:** The authors have declared that no competing interests exist.

control their everyday life. Support and follow-up from healthcare professionals, employers, family, and friends were crucial when learning to live with major LARS. Participants expressed desire for a systematic and proactive follow-up from healthcare professionals and contact with peer-support groups.

## Introduction

Colorectal cancer is a disease with poor prognosis [1]. Worldwide, it ranks third in terms of incidence but second in terms of mortality [1, 2]. Increasing age and socioeconomic development has been associated with colorectal cancer and since life expectancy in developing countries is improving, the incidence of colorectal cancer is predicted to increase to 2.5 million new cases worldwide by 2035 [1, 3].

Even though advances in radio- and in chemotherapy treatment have improved survival rates for patients diagnosed with colorectal cancer, surgery remains the gold standard in terms of curative treatment [1]. Low anterior resection of the rectum with total mesorectal excision allows patients with colorectal cancer to avoid a permanent colostomy by preserving the anal sphincter [1]. Still, this technique can also lead to challenging conditions such as low anterior resection syndrome (LARS) [4]. LARS has traditionally been defined as a disordered bowel function following rectal resection that leads to detriments in quality of life [4, 5]. LARS is characterised by altered bowel function with symptoms like faecal incontinence, urgency, increased frequency, fragmentation of stools, constipation, and incomplete emptying [6]. In recent years, a definition that includes symptoms and consequences of the condition has been proposed [7]. According to the International Consensus Definition of Low Anterior Resection Syndrome, LARS is present when a patient who has had an anterior resection (sphincter-preserving rectal resection), experiences at least one in eight symptoms that result in at least one of eight consequences [7], as shown in Table 1. Studies has shown that 60–90% of patients suffer from LARS even years after surgery [5, 6].

It has been suggested that the body and the world create a mutual connection, and when the body changes the world also changes [8]. Bodily intelligence, the skillful body and embodied knowledge are terms that have been used to describe how people living with chronic illnesses possess resources and develop strategies for health promotion and salutogenesis [9].

Next to surviving cancer, avoiding surgical complications and a permanent stoma are important issues for those undergoing colorectal surgery [10]. Yet, learning how to manage new bowel functions and alterations associated with LARS can represent an additional challenge

**Table 1. Symptoms and consequences that a patient with an anterior resection (sphincter-preserving rectal resection) can have, as suggested by the International Consensus Definition of Low Anterior Resection Syndrome.**

| Symptoms | Consequences |
|---|---|
| Variable, unpredictable bowel function | Toilet dependence |
| Altered stool consistency | Preocupation with bowel function |
| Increased stool frequency | Dissatisfaction with bowels |
| Repreated painful stools | Strategies and compromises |
| Empying difficulties | Impact on: |
| Urgency | Mental and emotional wellbeing |
| Incontinence | Social and daily activities |
| Soiling | Relationships and intimacy |
| | Roles, commitments and responsibilities |

for cancer survivors [11, 12]. Adequate pre- and postoperative follow-up is therefore of importance for healthcare providers striving to deliver support for patients so they can cope with the consequences of cancer surgery in general, and LARS in particular [13, 14]. A large part of the recovery process following colorectal cancer surgery occurs at home, where counselling from healthcare professionals is less available [14]. It is also in this environment the consequences of living with an unpredictable bowel function might start affecting patients' daily life [15, 16]. Even though increased focus on the symptoms and consequences of LARS have been warranted [7, 13], studies exploring the short-term experiences of patients' living with major LARS and how they manage bowel challenges are scarce. Currently, available studies describe the experiences of individuals who have been living with LARS one year or later after surgery [16, 17], leaving behind the initial, and perhaps more significant, months following hospital discharge. Therefore, the aim of the study is to explore patients' experiences of living with major LARS in the first three to six months after sphincter-preserving colorectal surgery.

## Materials and methods

### Design

We conducted a phenomenological qualitative study as proposed by Giorgi [18]. This approach describes the individuals' lived experiences of a phenomenon as depicted by the participants [19]. The study has followed the consolidated criteria for reporting qualitative research (COREQ) guidelines [20] and the Human Participants Research Checklist as suggested by PLOS One (see S1 and S2 Appendices).

### Participants and setting

We followed the guiding principles for selecting a sample in phenomenological studies stating that participants must have experienced the phenomenon to be investigated, and that they are able to articulate how it was to live with this phenomenon [21]. Participants were recruited from a public county hospital in Norway performing about 50 operations for colorectal cancer annually. Control appointments are regularly scheduled for these patients, three to six months after surgery. A clinical nurse specialist working at the outpatient clinic asked patients who fulfilled the inclusion criteria to participate in the study, and those consenting were contacted by the first author. Inclusion criteria were: 1) adult patients (>18 years) previously diagnosed with colorectal cancer, 2) having received sphincter-preserving low anterior resection surgery during the last 6 months, 3) experiencing major LARS after surgery, 4) able to understand and speak Norwegian and 5) able to provide written consent to participate in the study. Exclusion criteria were 1) patients undergoing palliative treatment, 2) patients with metastatic disease, and 3) other post-operative complications.

   Major LARS was identified through the validated LARS score, an instrument designed to assess patients' symptoms of LARS and the impact that these symptoms have on quality of life [22]. LARS score measures flatus incontinence, liquid stool incontinence, stool frequency, stool clustering and faecal urgency [22]. This instrument has shown good psychometric properties [23]. LARS scores range from 0 to 42 with high scores representing higher impact on patients' quality of life [22]. LARS scores can be divided into three categories: 0–20 (no LARS), 21–29 (minor LARS) and 30–42 (major LARS) [22].

### Data collection

Semi-structured interviews of included patients followed an interview guide consisting of open-ended questions. Questions were based on previous literature [1, 23, 24] and on clinical

**Table 2. Interview guide.**

| |
|---|
| 1. What is it like to live with low anterior resection syndrome? |
| 2. Can you tell me how your bowel function has been after you had colorectal cancer surgery? |
| 3. Can you say something about the information you received before surgery regarding what the effects on your bowl function might be after surgery? |
| 4. Can you say something about what kind of information you received in the discharge conversation with the doctor and nurse on the ward about what the period after surgery would be like. What advice did you get? |
| 5. Can you tell me about how bowel emptying challenges have affected your daily life? |
| 6. Can you tell me about how it has affected your relationships and sexuality? |
| 7. Can you tell me how you have chosen to deal with bowel emptying challenges? |
| 8. Can you tell me something about who you've been able to talk to about your bowel emptying challenges? |
| 9. Can you tell me something about how you have been treated by doctors and nurses when you have brought up the issue of your bowel emptying challenges? |
| 10. Can you say something about what follow-up you would like in relation to the challenges you have experienced? |
| 11. Can you talk about your thoughts on not having a stoma before surgery and what you think about that option today? |

experience (Table 2). User participation was ensured on several stages of designing the interview guide. First, involving a patient with major LARS who had received follow-up at the outpatient clinic and who provided valuable feedback regarding the interview guide. Second, receiving advice from another patient with major LARS who completed a pilot interview and who suggested to include questions related to relationships and sexuality.

In total, five interviews were conducted from September 2022 to January 2023. Interviews began with an open approach where the participants talked about how they experienced living with major LARS. The interviewer listened actively, kept eye-contact, nodded along as confirmation of interest and asked relevant questions about each participant's story so that the conversation proceeded naturally. Three interviews took place in the privacy of the participants' homes and two in a meeting room at the hospital, following participants own wishes. All participants gave consent to have the audio of their interviews recorded. Interviews lasted between 25 to 35 minutes and resulted in rich narratives.

## Data analysis

The phenomenon to be studied was "living with major LARS as a whole". Interviews were transcribed verbatim by the first author, and analysis proceeded simultaneously with transcription. The transcribed interviews were analysed using the four-step process of Giorgi's (2009) [18] phenomenological method. First, interviews were read repeatedly to achieve an immediate initial understanding of the phenomenon [18]. Second, meaning units were identified and marked. A meaning unit is a constellation of words or statements that relate to the same central meaning assumed by the researcher [18]. In the third step, meaning units are transformed from participants' natural expressions into the language of healthcare professionals [18]. Comparable meaning units were identified, and these formed the basis for the themes that emerged. Step four involved grouping comparable themes from each interview and synthesising the essential structures of all the transformed meaning units into consistent statements regarding participants' experiences of living with low anterior resection syndrome [18]. Table 3 shows an example of the analytical process.

## Ethical approval

The study was approved by the Norwegian Agency for Shared Services in Education and Research (ref: 849190), and conforms with the principles outlined in the Declaration of

**Table 3. Examples from the data analysis using Giorgi´s phenomenological method.**

| Meaning unit | Essential structures | Main theme |
|---|---|---|
| «Yes, they said that there would be periods of frequent and loose stools. But you can't, you don't realise anyway, you hear what they are saying, but you don't understand what it means. It is like having diarrhoea for several days in a row somehow, all the time" | Unable to imagine what it will be like, despite good information before surgery | You hear what they are saying, but don't understand what it means until your body goes through it. |
| "I got lots of information, but I don't think it was properly explained to me how it would turn out in reality" | Got a lot of information but a lack of information about what the bowel function was going to be like | |
| "Maybe a little more about what this could be like if there wasn't a stoma then. Then you get a challenge with your gut, right, so you are aware of it". | Want more information about how bowel function would be after surgery | |
| "The doctors say that I have to be patient and that it can still stabilize. And I guess that's what the nurses also have said, but nurses have also given good advice about what I can do. [Name of the medicines] and, yes, a bit like that . . . diet and uh, yes" | Useful advice from nurses about conservative measures. Both doctors and nurses say that it will stabilize. | |

Helsinki [25]. The data protection officer at the hospital and the head of the surgical department also approved the study. Participants signed a written consent form and were informed that participation in the study was voluntary, that the interviews would be treated confidentially, that they could withdraw from the study at any time and that data would be stored on a secure server at the hospital until the end of the study. All the participants were offered a follow-up consultation by a clinical nurse specialist after the interview if they felt a need to talk about options to improve their bowel challenges.

## Rigour

Several actions were taken to increase credibility, transferability, reflexivity and transparency, and to so strengthen trustworthiness [26]. By recruiting participants who had experienced the phenomenon to be studied, who were able to articulate how it was to live with LARS, and by having three researchers contributing to the data analysis, the credibility of the study was strengthened. Transferability was safeguarded by providing a description of the participants, the setting, the research process, by making participants' voices visible in the quotations and by discussing the findings in relation to other international studies. Employing the phenomenological attitude, or reduction, also referred to as bracketing, is fundamental for demonstrating rigour and reflexivity in descriptive phenomenology [18]. Reflexivity refers to the process of critical self-reflection about oneself as a researcher and one's influence on the research process [26]. The first author conducted all the interviews and had no treatment relationship with the participants. Because the first author had previous experience in counselling patients with LARS, a reflexive journal was maintained throughout the study to capture pre-study beliefs and frames of reference beforehand in order to reduce prejudice on the data. To enhance transparency and reflexivity, the interview guide, the data collection method, and the data analysis were discussed between authors.

As suggested by Malterud et al. [26], we used the concept of information power when assessing the number of participants. According to this concept, the more information relevant for the actual study the sample holds, the lower the number of participants needed [26].

## Results

Five interviews, three of them with women, were performed. The mean age of the participants was 67 years, all were married and were living in heterosexual relationships. Participants had major LARS at inclusion with scores ranging between 30–42, had anastomoses of 5–10 cm and none of them had had a temporary stoma. There was some variability in the treatments received as three participants were managed by surgery alone, one had neoadjuvant radio-chemotherapy and one had adjuvant chemotherapy. The two participants who had undergone neoadjuvant or adjuvant chemotherapy had the highest LARS score (Table 4).

Patients' experiences when adapting to living with LARS could be grouped into five themes, as shown in Fig 1.

### Theme 1. Bowel emptying becomes all-consuming and affects both daily life and working life

Complications after treatment affected participants' life in different ways. Issues they did not think about before, like going to the toilet or how their bowel functioned, were now a concern. Unlike before cancer treatment, participants reported having to think about the location of toilets when they took part in activities outside their homes. Bowel emptying took place immediately in the morning or after meals. Some participants described going to the toilet five times in a lapse of 30 minutes, other participants described using the toilet between five and thirty times a day. All participants reported bowel movements associated with rectal urgency, and when the urge to defecate occurred, they had to rush to the toilet.

> *When I first have to go to the toilet, then I really have to go to the toilet, it's not possible to wait too long [. . .] when it comes, then I have to. . . then I have to fly.*

Participants described the initial post-operative period as unpredictable and demanding, with increased stool frequency and loss of control over their bowel function. However, they also described daily variations.

**Table 4. Characteristics of patients included in the study (N = 5).**

|  | Mean or count |
|---|---|
| Age (range) | 67.4 (56–81) |
| Gender (Female) | 3 |
| Marital status (married) | 5 |
| Education level | |
| Elementary school | 1 |
| High school | 2 |
| University | 2 |
| Work status | |
| Part-time | 2 |
| Retired | 3 |
| Type of surgery | |
| PME | 3 |
| TME | 2 |
| Anastomose level in cm (range) | 8 (5–10) |
| LARS score at 3 months follow-up (range) | 37.6 (32–41) |

LARS = Low anterior resection syndrome; PTM = Partial mesorectal excision; TME = Total mesorectal excision

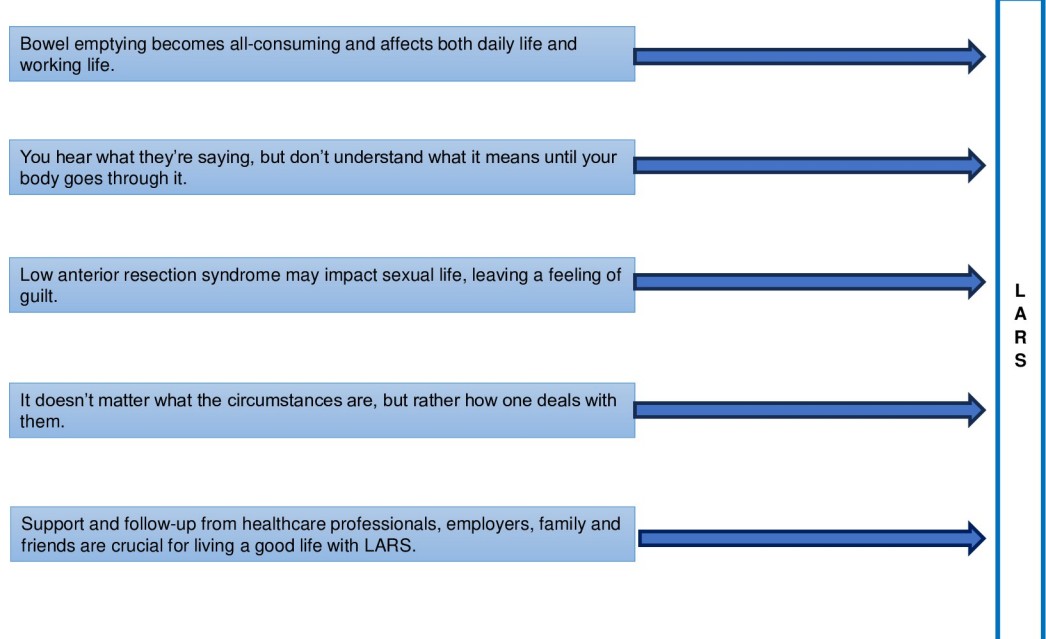

**Fig 1. Main themes related to patients' experiences of living with major LARS in the first three to six months after surgery.**

> *I can have days where I'm not on the toilet and I can have days where I'm on the toilet ten to fifteen times. When I have such a proper rinse, that is, then I can sit on the toilet for quite a long time. Then it's a lot of work, then I notice it's fierce.*

As bowel function gradually improved, participants became accustomed to the challenges. Yet, they still had to be aware about the location and access to toilets when outdoors.

> *I've gotten used to it, so I just make sure I'm somewhere near a toilet and have a clear path to run down. . . So, when I've been out to eat, it has been good to know where it is, but it has gone well.*

Participants perceived bowel emptying as an all-consuming task that required precautions on everyday life. A participant created daily routines before she could take a walk with the family's dog.

> *You don't take the longest walks with the dog; they are just short walks. I must finish going to the toilet in the morning, preferably 2–3 times. It's rare that something happens before 11 a.m. Can't go on a trip until I've finished emptying, bowel emptying challenges become all-consuming.*

However, participants eventually overcame bowel challenges. They learned how to live a more normal life by making good plans.

> *I live an ordinary life. I've been on summer holiday, we've been on boat trips. Yes, yes, it's fine, but I need to know where the toilets are. Plan what I am going to do.*

Two participants had returned to work, but none of them were working full-time because of their variable and unpredictable bowel function. By the time interviews were conducted, one of these participants had reassumed 40% of the working duties, while another was able to perform 60% of previous work.

*It was too much to work three days. Because I get very tired going to the toilet all the time. So, I think maybe the mental stress takes it out of you pretty much after what you've been through. . . So, of course it affects work, but day-to-day life goes smoothly at home I think, as I am living pretty normally.*

## Theme 2. You hear what they're saying, but don't understand what it means until your body goes through it

All participants remember receiving information about LARS. However, it was difficult to picture the radical consequences cancer treatment would bring to their life. Participants were mostly satisfied with the information received before surgery but wished having more knowledge about how intense the symptoms of LARS would be and what to expect after hospital discharge.

*Yes, they said that there would be periods of frequent and loose stools. But you can't, you don't realise anyway, you hear what they're saying, but you don't understand what it means. It's like having diarrhoea for several days in a row somehow, all the time. I got lots of information, but I don't think it was properly explained to me how it would turn out in reality.*

Participants did not recall receiving advice on how to understand their new bowel function before hospital discharge. One of them was told that everything was working fine, that he/she needed to be patient and take the time needed to recover. Another participant described receiving good information about the physical aspects of the body after surgery, but information about the psychosocial consequences of LARS were lacking.

Several participants were satisfied with the follow-up consultation provided by the clinical nurse specialist. They remembered being allowed to ask questions regarding LARS and receiving useful advice linked to bowel emptying issues. One of the participants also benefited from follow-up by a physiotherapist and by doing pelvic floor training.

Participants conveyed that it was also important to tell future patients that the challenges associated with LARS could get better. One participant, for example, experienced that her bowel function improved after some months and believed it was valuable information to other patients with LARS.

*The doctors say that I have to be patient and that it can still stabilise. And I guess that's what the nurses also have said, but nurses have also given good advice about what I can do. [Name of medicines] and yes, a bit like that. . .diet and uh, yes.*

## Theme 3. Low anterior resection syndrome may impact sexual life, leaving a feeling of guilt

Some participants described changes in intimacy after surgery. One of them expressed having resumed sexual life, one had put sexual life on hold while others stated not having a sexual life at all. Fear of faecal leakage during sexual intercourse was a major concern for one of the

participants. This informant described how, at first, it was completely out of the question because she was terrified that stool would slip out.

The participant who had put her sexual life on hold had been married for a long time, and according to her, sexual intercourse was not a priority at the present time. Despite having a husband who was understanding, she felt guilt about putting their sexual life on hold. For participants describing not having a sexual life, this had also been the case before surgery. For some of those, sexual intercourse was replaced with closeness and with other forms of intimacy. They explained that sexuality to them was about so much more than sexual intercourse.

*I have a husband who is kind and helpful and cares about me, and we have an equal and good relationship, then I think that maybe it's just as important at my age as having a very exiting sexual life. To hug each other, be nice to each other in other ways, to stroke each other. . . And the hugs in the morning and the goodnight kisses, or whatever it is, it matters a lot. Being able to lie in each other's arms or yes. . . Yes, that closeness may mean just as much. . .*

## Theme 4. It doesn't matter what the circumstances are, but rather how one deals with them

Participants tried to find strategies to cope with LARS. They used approaches which primarily involved taking medications for diarrhoea and/or constipation, adapting their diet, taking probiotics and using pantyliners. Dietary changes could involve completely refraining from eating or avoiding certain food. One of the participants who was back at work explained that at home she ate as normal, but at work she did not eat at all to prevent having to go to the toilet constantly. Another participant had to stop eating spicy food, which he loved.

*We had such a strong chilli stew here one day, and it was two days before I was normal again. So, I try to stay away from it. It's a shame because I love spicy food.*

All participants displayed a positive attitude in resuming everyday life after colorectal cancer surgery. They lived life as normally as they could, with some adjustments, and tried not to worry too much. By this stage, they had come to accept their situation, despite the issues related to LARS. One of the participants made it clear that there were hard days in-between but had a positive attitude since the prognosis was good and because there were others who had it worse. This participant talked about patients living with Crohn's disease and reflected on the fact of keeping things in perspective.

Participants described taking precautions by going to the toilet before taking a walk or using pantyliners or diapers to avoid soling their clothes. One participant used toilet paper, putting three or four sheets together in the underwear as a precaution.

*I think it's okay to go to the toilet two times in the morning before I go for a long walk. But I was out for four hours yesterday, and it went just fine. So, it worked out. I use pantyliners just in case.*

Not having control over flatulence was commented by several participants. One of them, working in a kindergarten, decided not to care about it.

*If there is a fart, then there will be a fart. Not the worst thing happening in this world, so it's fine. Because it does happen from time to time. . .the kids at work think it's funny. Yes, but*

*that's how it can be. Without me noticing it's coming. Hmm, it can do that. I've decided not to care about it.*

Despite the negative effects of LARS, all participants preferred their current bowel pattern rather than a life with a stoma. However, they would accept a stoma if medically necessary.

*It is possible to live a full life with low anterior resection syndrome. Rather that, than a stoma bag on the belly. I think I'd have a real challenge having to use one.*

### Theme 5. Support and follow-up from healthcare professionals, employers, family and friends are crucial for living a good life with LARS

All participants recognised the importance of support and follow-up from healthcare professionals, employers, family and friends to live a good life with major LARS. Talking openly about their struggles provided support and eased their worries. One participant spoke about sharing her experiences at work and with friends. She had been on cabin trips and to festivals, and everyone knew what she had been through because she did not bother hiding it. Another participant who was active in volunteer work chose to be open about her situation.

*They [the volunteer network group] gave me flowers when I was sick. So, I wanted to say something and thank them at a meeting, there were about 40–50 people present, and I thanked them and said a little bit about my situation. And I've only experienced positive responses and feedback afterwards. Because people think it's good that I'm open [about having major LARS] and I experienced someone coming up to me afterwards and saying that they themselves have some of the same challenges. So, I think it's a good idea to be open. I lose nothing from doing so myself, and if I think I can help someone else&*

Participants who were back in work had a good working relationship with their employers and described them as good facilitators. One of the participants who worked in a health care facility experienced positive dialogues with her employer. Arrangements were made for her to continue working because it often involved heavy lifting.

*I have had a good relationship with my job. Eh, I would wait until, until I was sure, until I felt like I could handle going to work and not be afraid of my bowl function, because I work in [place of work], a heavy somatic ward, and there's a lot of lifting. Even though we have assistive devices and elevators and everything like that, you must get them higher in bed manually, it doesn't happen by itself. So, I wanted to feel like I was able to help, without pooping myself, to put it bluntly.*

Participants experienced that follow-up from healthcare professionals went smoothly. They followed the clinical pathway designed for cancer treatment and felt that their five-year follow-up plan was reassuring.

*Once I got onto that clinical pathway, I had fantastic follow-up. I have to say that. Doctors and nurses have only been nice and positive, no bad experiences with anything. I also have a general practitioner who is very good, who I can talk to and who supports me. I send him messages on "Health Norway," a digital platform, and he answers regardless of whether it's a weekday or the weekend.*

One of the participants wished she had been able to talk to a stoma care nurse before it was time for the three-month's follow-up control. She experienced receiving important information about her diet, antidiarrheal medication and about incontinence skincare that could have been useful in an early stage. Participants experienced that they had good conversations with the stoma nurse and they felt free to ask questions.

*Perhaps I could have talked to the stoma care nurse earlier, just before you're done with everything. She was the last person I spoke to at the hospital. She could perhaps have been one of the first. So, then I could have had [name of a medicine for diarrhoea] and taken it several months before I actually got it. Also, with the [name of the product] cream and all the things there could have been passed on to me much earlier.*

One of the participants expressed a desire to talk with peers and others who also had experienced major LARS and considered this would be helpful. Peers would be able to understand the situation better than a person who had not been in the same situation. This participant suggested having small networking groups for people living with LARS since it could sometimes be difficult.

## Discussion

This study provides insight into patients' experiences living with major LARS during the first three-to six months following colorectal cancer surgery by using a phenomenological approach [27]. Participants describe learning how to live with a bowel function that is variable and unpredictable, especially in the early months following surgery. According to their descriptions, the symptoms of major LARS forced them to always have to be in control of where they were in relation to the toilet. Participants describe having to learn coping strategies, such as taking antidiarrheal medication, adapting their diet, and protecting their underwear from soiling with diapers or pantyliners in order to deal with the consequences of major LARS.

The results of our study support the novel International Consensus Definition of Low Anterior Resection Syndrome [7]. Our participants describe struggles with an altered bowel function and LARS symptoms affecting daily- and working life. According to Merleau-Ponty [8], it is through our physicality that we are in the world and have access to it and to ourselves. He considers being a subject as being in the world as an experienced body, and when something about the body is altered, it will lead to a change in the experienced body [8]. Transferred to our study, this means that the symptoms of major LARS bring consequences on how the world is experienced and handled by our participants. The variable and unpredictable bowel function leads to the toilet having a more central place in their life than before and have an impact on other social- and mental areas.

Another phenomenological study [28] describes that participants experienced the initial lack of bowel control as to "living in the restroom", but gradually developing coping strategies. Coping strategies have also been highlighted by the International Consensus Definition of Low Anterior Resection Syndrome [7]. Early in the rehabilitation process, our participants seem to have developed personal knowledge about their new body. Such bodily knowledge helps us to assess important values in life and to focus on opportunities [9]. Our participants are mature people, with a vast life experience, and who seem to have good support in family and friends. These assets might help them in finding strategies to cope with the challenges that major LARS brings and to help them live a relatively normal life as they resume work and leisure activities. Resilience is viewed as an important way of coping, and it is through resilience that patients recover from or avoid negative outcomes from burdensome conditions [29].

Our participants also expressed desire to join peer support groups, as meeting someone with the same challenges was viewed as useful. These findings are consistent with others [17, 30], who reported adaption to LARS by being open about the condition and by getting support from their social environment. Our findings are important because our patients are reporting these coping strategies already three to six months after colorectal surgery, as opposed to one- and three years later [15, 17, 30]. This information is valuable for healthcare providers wanting to provide support for patients experiencing major LARS and for their relatives.

All our participants reported improvements in their bowel function as time went by. It has been reported that the severity of LARS symptoms can diminish six months after surgery and that bowel function can continue to improve during the first two years [31]. However, other studies show that patients still experience LARS several years after completion of therapy [5]. Our participants highlighted the importance of receiving realistic information from healthcare professionals that also convey that bowel function could improve. This information could bring hope for recovery to prospective patients. Hope and optimism are important for coping and resistance to illness [31]. According to Reinwalds et al. [16], patients with LARS will reach a point where they decide to accept the challenges of an altered bowel function. Problem-focused and emotion-focused coping strategies can help patient with major LARS adapting to the stressors they are exposed [32]. Bowel dysfunctions associated with major LARS may be defined as stressors, and in this study the participants used problem-focused and emotion-focused coping strategies. Problem-focused strategies, such as taking medication, adjusting their diet, exercising their pelvic floor and undertaking physiotherapy, reduced their bowel challenges. In relation to emotion-focused strategies, one of the participants tried to take into consideration other patient groups and their challenging situations, thereby putting their own experience into a wider perspective.

The International Consensus Definition for LARS addresses relationships and intimacy [7]. According to this definition the consequences of LARS on sexuality are not only due to changes in sexual function but are also related to its impact on intimacy [7]. Our informants have a vast life experience, have been married for a long time and have good support in both family and friends. These assets might have contributed to develop coping strategies to deal with the challenges major LARS brings, and to live a relatively normal life as they resume work and leisure activities. One of the participants emphasized that closeness and care from a spouse meant as much as having sexual intercourse. Our findings are also in accordance with a phenomenological study, reporting that participants changed intimacy behaviours in order to express love [30]. The World Health Organization defines sexuality as more than sexual intercourse [33]. Sexuality is "a central aspect of being human throughout life encompasses sex, gender identities and roles, sexual orientation, eroticism, pleasure, intimacy and reproduction. Sexuality is experienced and expressed in thoughts, fantasies, desires, beliefs, attitudes, values, behaviours, practices, roles and relationships" [33]. One of the participants felt guilt and worried about no longer meeting her partners sexual needs. Feelings of guilt are also described by others [13] as fear of leakage and flatulence affected participants sexual life. Patients with major LARS are in a vulnerable situation where sexual problems require support and understanding from their partners. Although an altered bowel function is the major post-operative problem in colorectal cancer patients, urinary and sexual problems are common for men and women after treatment for colorectal cancer, and these issues influence patients' intimacy with their partners [28]. Thus, sexual health is an important subject for nurses to discuss with patients suffering from LARS.

Follow-up consultations after colorectal surgery in our hospital are performed after approximately three months. Even though our participants were satisfied with follow-up from

healthcare professionals, they also reported missing having contact with healthcare professionals in the time period from hospital discharge to the first follow-up consultation time. It has been suggested to implement proactive counselling strategies for patients with major LARS both early in the treatment trajectory and in the longer run [13, 34]. In the Danish sequelae clinics, patients with LARS are offered follow-up by a nurse-led clinic that gives advice on dietary and medical treatment of diarrhoea or constipation, on the use of bulk-forming supplements and in transanal irrigation [35]. A study by Pape et. al (2021) found that patients experienced that adjusting their diet, taking medication and using incontinence care products had an effect on LARS, and that other treatment options such as physiotherapy, transanal irrigation and sacral nerve stimulation were not used by the majority of patients [5]. Van der Heijden et al. [14] recommends healthcare professionals to make phone contact with patients in an early stage after hospital discharge to offer support and practical advice. We believe that this kind of contact is important for our patients also in the period of time from hospital discharge to the follow-up appointment three months later, to prevent loneliness and support hope in the patient's trajectory [36], as well as alleviate the partners' needs [37].

Our findings indicate that stoma care nurses play an important role educating patients with LARS, helping them to cope with altered bowel function and teaching how to live a good everyday life with the condition. Yet, stoma care nurses could also take a more proactive role by asking questions and making inquiries related to LARS as it has been reported that some patients find it uncomfortable talking about those issues [15, 38]. Nurses can also help patients connecting to peer support groups if available.

## Strengths and limitations

Among the strengths of our study is the use of a phenomenological approach congruent with the aim of the study: exploring patients' early experiences of living with major LARS. Another strength lies in the fact that patients currently living with LARS actively helped developing the interview guide, assuring therefore a patient perspective in research. Besides, a pilot interview was performed with a second patient and the valuable input from this informer contributed to include questions that increased the flow of information, and which could in turn improve health approaches to patients living with major LARS.

Some limitations must also be mentioned. Participants were recruited from one public county hospital in Norway. Yet, this hospital is responsible for providing advanced health services, including cancer surgery, for a population of 250 000 inhabitants, living in six different municipalities of the country [39]. Since Norway has some of the highest incidence and mortality rates of colorectal cancer in the world [40], we believe that we were able to capture how patients experience living with LARS in the early stages of the condition. It can be argued that our sample size was small, which may be explained by the number of patients operated annually in this hospital and by only enrolling patients with sphincter preserving surgery without having an temporary ileostomy during their disease trajectory. However, we have followed the principles for selecting a sample in phenomenological studies [21] and taken into consideration that phenomenological studies include a range of patients between 3 and 10 [19]. Our sample was fairly homogeneous including only married participants living in heterosexual relationships, which may impact the transferability of the results. A sample including other age groups, a different gender distribution, different ethnic backgrounds and different sexual orientations might have provided additional insights. However, to strengthen transferability, we have provided a description of the participants, the research process, and made participants voices visible in the results [26].

## Conclusion

This study provides insight into the participants experiences of living with major LARS the early months following colorectal cancer sphincter-preserving surgery. All participants described some form of dependency on the toilet, especially in the initial period after hospital discharge. Still, they also developed strategies that enabled them to control their everyday life and made them able to live a fulfilling life with major LARS. Support and follow-up from healthcare professionals, employers, family, and friends were crucial to live with major LARS. However, there was a desire for a systematic and proactive follow-up from healthcare professionals and contact with peers. The role of the clinical nurse specialist was viewed as vital by giving advice to help patients to live a better life with major LARS.

## Supporting information

**S1 Appendix. COREQ (COnsolidated criteria for REporting Qualitative research) checklist.**
(DOCX)

**S2 Appendix. Human participants research checklist.**
(DOCX)

## Acknowledgments

We would like to thank stoma care nurse Torill Elin Olsen and the two patient representants for valuable input in the development of the interview guide.

## Author Contributions

**Conceptualization:** Camilla Løvall, Marit Hegg Reime.

**Data curation:** Camilla Løvall.

**Formal analysis:** Lotte Miriam Eri Mjelde.

**Methodology:** Camilla Løvall, Marit Hegg Reime.

**Project administration:** Camilla Løvall.

**Supervision:** Leslie S. P. Eide, Marit Hegg Reime.

**Writing – original draft:** Camilla Løvall.

**Writing – review & editing:** Lotte Miriam Eri Mjelde, Leslie S. P. Eide, Marit Hegg Reime.

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
