## [Decision Letter · Decision Letter 0]

1 Apr 2024

PONE-D-23-42953Patients’ experiences of living with low anterior resection syndrome three to six months after colorectal cancer surgery: A phenomenological studyPLOS ONE

Dear Dr. Reime,

Thank you for submitting your manuscript to PLOS ONE. After careful consideration, we feel that it has merit but does not fully meet PLOS ONE’s publication criteria as it currently stands. Therefore, we invite you to submit a revised version of the manuscript that addresses the points raised during the review process.

We look forward to receiving your revised manuscript.

Kind regards,

Atalel Fentahun Awedew, MD,MPH, Ass.Prof Surgery

Academic Editor

PLOS ONE

Journal Requirements:

2. We note that Figure 1 in your submission contain copyrighted images. All PLOS content is published under the Creative Commons Attribution License (CC BY 4.0), which means that the manuscript, images, and Supporting Information files will be freely available online, and any third party is permitted to access, download, copy, distribute, and use these materials in any way, even commercially, with proper attribution. For more information, see our copyright guidelines: http://journals.plos.org/plosone/s/licenses-and-copyright.

(1) You may seek permission from the original copyright holder of Figure 1 to publish the content specifically under the CC BY 4.0 license. 

3. In this instance it seems there may be acceptable restrictions in place that prevent the public sharing of your minimal data. However, in line with our goal of ensuring long-term data availability to all interested researchers, PLOS’ Data Policy states that authors cannot be the sole named individuals responsible for ensuring data access (http://journals.plos.org/plosone/s/data-availability#loc-acceptable-data-sharing-methods).

**Additional Editor Comments:**

Your made interview only five patients. Why do you make small sample?

Align your manuscript with PLOS ONE guideline

Reviewers' comments:

Reviewer's Responses to Questions

**Comments to the Author**

1. Is the manuscript technically sound, and do the data support the conclusions?

Reviewer #1: Yes

Reviewer #2: Yes

2. Has the statistical analysis been performed appropriately and rigorously? 

Reviewer #1: Yes

Reviewer #2: Yes

3. Have the authors made all data underlying the findings in their manuscript fully available?

Reviewer #1: Yes

Reviewer #2: Yes

4. Is the manuscript presented in an intelligible fashion and written in standard English?

Reviewer #1: Yes

Reviewer #2: Yes

5. Review Comments to the Author

Reviewer #1: It is an interesting area of research undertaken by your team gathering insight into living with LARS. My only concern is why the researchers have settled with 5 participants only to give their recommendations. Was it not possible to recruit more patients to have a broader inclusion and diverse opinions?

The description of the findings is appropriate in a good sequence.

Apart from this limitation of a small sample. this study could be helpful in stimulating more research in the same area.

Reviewer #2: Strengths and weaknesses

1. the topic is not unique but worthy of researching

2. there are of papers in google scholar and Refseek about this topic since 2020

3. Ethical approval is mentioned

4. the title is too long. You can shorten it to be more attractive and citable

5. the abstract contains plagiarism. Please revise it. Otherwise, it is informative

6. the aim is clear

7. the KEYWORDS are fair. Please use http://www.ncbi.nlm.nih.gov/mesh

8. lack of the abbreviations section

9. the introduction provide sufficient background information for readers in the immediate field to understand the problem/hypotheses

10. the text arrangement is good

11. the method section is good

12. the depth of the academic material is good

13. the study design is good

14. The suitability and accuracy of questions is good

15. The research methodology is suitable

16. The materials are good

17. the logic is clear

18. the paper is not novel

19. there are few grammatical errors in this article

20. the related concepts are introduced

21. the readability is sufficient

22. the results are good

23. all figures/tables are clear enough to summarize the results for presentation to the readers

24. all figures/tables are well referred to in the text

25. the theoretical analysis in this article is sufficient

26. the discussion of results from multiple angles is sufficient

27. the conclusion is tenable

28. the reference section contains too many old references

29. please use (google scholar and Refseek) search engines then set it since 2020

30. the references are in order within the text

31. Bias is present

32. There is no conflict of interest with the author about this topic

33. Fund is mentioned

34. Conflict of interest is mentioned

35. Acknowledgement is mentioned

36. You can use my suggestions

My final decision is acceptable after minor revision

6. PLOS authors have the option to publish the peer review history of their article (what does this mean?). If published, this will include your full peer review and any attached files.

Reviewer #1: **Yes: **Mohammed Amir

Reviewer #2: **Yes: **hazim alhiti

---

## [Author Response · Author response to Decision Letter 0]

3 May 2024

We have added a 12 page file with respons to editor and reviewers comments.

---

## [Editor Report · Decision Letter 1]

10 May 2024

PONE-D-23-42953R1Patients’ experiences of living with low anterior resection syndrome three to six months after colorectal cancer surgery: A phenomenological studyPLOS ONE

Dear Dr. Reime,

Thank you for submitting your manuscript to PLOS ONE. After careful consideration, we feel that it has merit but does not fully meet PLOS ONE’s publication criteria as it currently stands. Therefore, we invite you to submit a revised version of the manuscript that addresses the points raised during the review process.

**I extend my heartfelt appreciation to the esteemed authors for addressing all insightful comments and constructive feedback. The manuscript not only sheds light on sensitive issues in Colorectal surgery but also introduces a fresh perspective to the field. I have duly noted the minor formatting suggestions provided, including the incorporation of the study design in the methodology section and changing "aim" to "Objective." Following the suggested format of Background, Objective, Method, Results, Conclusion, and Keywords will enhance the clarity and structure of the manuscript. Please submit the clear manuscript with recommended format of abstract within 48hours** Please submit your revised manuscript by Jun 24 2024 11:59PM. If you will need more time than this to complete your revisions, please reply to this message or contact the journal office at plosone@plos.org. Please include the following items when submitting your revised manuscript:A rebuttal letter that responds to each point raised by the academic editor and reviewer(s). You should upload this letter as a separate file labeled 'Response to Reviewers'.A marked-up copy of your manuscript that highlights changes made to the original version. You should upload this as a separate file labeled 'Revised Manuscript with Track Changes'.An unmarked version of your revised paper without tracked changes. You should upload this as a separate file labeled 'Manuscript'.If applicable, we recommend that you deposit your laboratory protocols in protocols.io to enhance the reproducibility of your results. Protocols.io assigns your protocol its own identifier (DOI) so that it can be cited independently in the future. For instructions see: https://journals.plos.org/plosone/s/submission-guidelines#loc-laboratory-protocols. Additionally, PLOS ONE offers an option for publishing peer-reviewed Lab Protocol articles, which describe protocols hosted on protocols.io. Read more information on sharing protocols at https://plos.org/protocols?utm_medium=editorial-email&utm_source=authorletters&utm_campaign=protocols.

We look forward to receiving your revised manuscript.

Kind regards,

Atalel Fentahun Awedew, MD,MPH, Ass.Prof Surgery

Academic Editor

PLOS ONE
---

## [Author Response · Author response to Decision Letter 1]

23 May 2024

Mail correspondance with academic editor clarified that it was only the heading in the abstract we should edit. 

We have now revised the heading to Objective in the abstract, according to the following feedback: 

I extend my heartfelt appreciation to the esteemed authors for addressing all insightful comments and constructive feedback. The manuscript not only sheds light on sensitive issues in Colorectal surgery but also introduces a fresh perspective to the field. I have duly noted the minor formatting suggestions provided, including the incorporation of the study design in the methodology section and changing "aim" to "Objective." Following the suggested format of Background, Objective, Method, Results, Conclusion, and Keywords will enhance the clarity and structure of the manuscript. Please submit the clear manuscript with recommended format of abstract within 48hours.

We are thankful to the Academic Editor for responding to our email request, and hope that our respons are to your satisfaction.

---

## [Editor Report · Decision Letter 2]

27 May 2024

Patients’ experiences of living with low anterior resection syndrome three to six months after colorectal cancer surgery: A phenomenological study

PONE-D-23-42953R2

Dear Prof. Marit Hegg Reime

We’re pleased to inform you that your manuscript has been judged scientifically suitable for publication and will be formally accepted for publication once it meets all outstanding technical requirements.

Kind regards,

Atalel Fentahun Awedew, MD,MPH, Ass.Prof Surgery

Academic Editor

PLOS ONE
---

## [Editor Report · Acceptance letter]

3 Jun 2024

PONE-D-23-42953R2 

PLOS ONE

Dear Dr. Reime, 

I'm pleased to inform you that your manuscript has been deemed suitable for publication in PLOS ONE. Congratulations! Your manuscript is now being handed over to our production team.

Kind regards, 

on behalf of

Dr. Atalel Fentahun Awedew 

Academic Editor

PLOS ONE